# Approximating 1-Wasserstein Distance with Trees

**Makoto Yamada**                                                    *makoto.yamada@oist.jp*
*Okinawa Institute of Science and Technology*
*Kyoto University*
*RIKEN AIP*

**Yuki Takezawa**                                    *yuki-takezawa@ml.ist.i.kyoto-u.ac.jp*
*Kyoto University*
*RIKEN AIP*

**Ryoma Sato**                                            *r.sato@ml.ist.i.kyoto-u.ac.jp*
*Kyoto University*
*RIKEN AIP*

**Han Bao**                                                          *bao@i.kyoto-u.ac.jp*
*Kyoto University*

**Zornitsa Kozareva**                                               *zornitsa@kozareva.com*
*Meta AI*

**Sujith Ravi**                                                       *sravi@sravi.org*
*SliceX AI*

**Reviewed on OpenReview:** *https://openreview.net/forum?id=Ig82l87ZVU*

## Abstract

The Wasserstein distance, which measures the discrepancy between distributions, shows efficacy in various types of natural language processing and computer vision applications. One of the challenges in estimating the Wasserstein distance is that it is computationally expensive and does not scale well for many distribution-comparison tasks. In this study, we aim to approximate the 1-Wasserstein distance by the tree-Wasserstein distance (TWD), where the TWD is a 1-Wasserstein distance with tree-based embedding that can be computed in linear time with respect to the number of nodes on a tree. More specifically, we propose a simple yet efficient L1-regularized approach for learning the weights of edges in a tree. To this end, we first demonstrate that the 1-Wasserstein approximation problem can be formulated as a distance approximation problem using the shortest path distance on a tree. We then show that the shortest path distance can be represented by a linear model and formulated as a Lasso-based regression problem. Owing to the convex formulation, we can efficiently obtain a globally optimal solution. We also propose a tree-sliced variant of these methods. Through experiments, we demonstrate that the TWD can accurately approximate the original 1-Wasserstein distance by using the weight estimation technique. Our code can be found in `https://github.com/oist/treeOT`.

## 1 Introduction

The Wasserstein distance, which is an optimal transport (OT)-based distance, measures the discrepancy between two distributions and is widely used in natural language processing (NLP) and computer vision (CV) applications. For example, measuring the similarity between documents is a fundamental natural language processing task that can be used for semantic textual similarity (STS) tasks (Yokoi et al., 2020).

For CV tasks, because matches between samples using OT can be obtained, it is used to determine the correspondence between two sets of local features (Sarlin et al., 2020).

One of the most widely used applications of the Wasserstein distance is the word mover's distance (WMD) (Kusner et al., 2015), which measures the similarity between documents by solving the optimal transport problem. Recently, Yokoi et al. (2020) proposed a more effective similarity measure based on the word rotator's distance (WRD). The WMD and WRD are distance-based optimal transport metrics that can be estimated by linear programming with a $O(n^3)$ order of computation, where $n$ is the number of words in a document. Thus, using the WMD and WRD on a large number of documents is challenging.

A well-known speedup technique for OT problems is the use of the Sinkhorn algorithm (Cuturi, 2013), which solves the entropic regularized optimal transport problem. Using the Sinkhorn algorithm, we can estimate the WMD and WRD with a computational cost of $O(n^2)$. However, this is slow for most NLP and CV tasks. Instead of using linear programming and the Sinkhorn algorithm, we can speed up the computation by projecting word vectors into a 1D space and solving the OT problem. This approach gives the sliced Wasserstein distance (Rabin et al., 2011; Kolouri et al., 2016) and its computational complexity is $O(n \log n)$. Note that the result of the sliced Wasserstein distance can be different if we use another projection matrix. Thus, the performance of the sliced Wasserstein distance is highly dependent on the projection matrix. To alleviate this problem, we averaged sliced Wasserstein distances using different projection matrices.

The tree-Wasserstein distance (TWD) (Evans & Matsen, 2012; Le et al., 2019) computes the optimal transport on a tree, and it can be computed in $O(N)$, where $N$ is the number of nodes in a tree. Specifically, we first embedded word vectors into a tree and computed the optimal transport based on a tree metric. Because we can obtain an analytical solution to the OT problem on a tree, it can be efficiently solved. Moreover, the TWD includes the sliced Wasserstein as a special case and empirically outperforms the sliced Wasserstein distance (Le et al., 2019). The performance of TWD is highly dependent on the tree construction. One of the widely used tree construction methods is based on QuadTrees (Indyk & Thaper, 2003). Another approach uses a clustering-based tree algorithm (Le et al., 2019). Recently, Takezawa et al. (2021) proposed a continuous tree-construction approach. Although tree-based methods improve computational complexity by a large margin, existing tree-based methods do not satisfactorily approximate the Wasserstein distance with arbitrary metric.

In this study, we aimed to accurately approximate the 1-Wasserstein distance using the TWD. Specifically, we propose a simple yet efficient sparse-learning-based approach to train the edge weights of a tree. To this end, we first demonstrate that the 1-Wasserstein approximation problem can be formulated as a distance approximation problem by using the shortest path distance on a tree. We then show that the shortest path distance can be represented by a linear model when we fix the tree structure. The weight estimation problem is then formulated as a non-negative Lasso-based optimization problem. Owing to the convex formulation, we can efficiently obtain a globally optimal solution. Moreover, we propose a tree-sliced variant of the proposed method. One of the key advantages of our formulation is that the optimization can be efficiently solved using an off-the-shelf solver. Through experiments, we evaluated the proposed algorithm using Twitter, BBCSport, and Amazon datasets and showed that the 1-Wasserstein distance computed by linear programming can be accurately approximated by the TWD, whereas the QuadTree (Indyk & Thaper, 2003) and ClusterTree (Le et al., 2019) have high approximation errors.

**Contribution:** The contributions of our paper are summarized as follows:

- A Lasso-based weight estimation is proposed for the TWD.

- An estimation of the weights for the tree-sliced Wasserstein distance is proposed.

- We empirically demonstrate that the proposed method is capable of approximating the 1-Wasserstein distance accurately.

## 2 Related Studies

**Optimal Transport and Wasserstein distance:** Measuring the distance between sets is an important research topic in machine learning, computer vision, and natural language processing. In early studies, the earth mover's distance (EMD), which can be computed using linear programming, was proposed as an optimal transport-based distance, equivalent to the Wasserstein distance, for cases where distance was employed to compute the cost function. The EMD has been particularly studied in the computer vision community because it can be used to obtain matching between samples. For example, Sarlin et al. (2020) proposed the SuperGlue method, which determines the correspondence between two sets of local features by using optimal transport. Liu et al. (2020) used the optimal transport method for semantic correspondence. For NLP tasks, Kusner et al. (2015) proposed the word mover's distance (WMD)and were the first to use the Wasserstein distance for textual similarity tasks; this distance is widely used in NLP tasks, including text generation evaluation (Zhao et al., 2019). Recently, Sato et al. (2022) re-evaluated the WMD in various experimental setups and summarized its advantages and disadvantages. Another promising approach is based on the word rotater's distance (WRD) (Yokoi et al., 2020), which normalizes the word vectors and solves the WMD with an adjusted probability mass. It has been reported that its performance is significantly better than that of the WMD. However, the computational cost of these approaches is very high. Therefore, they cannot be used for large-scale distribution-comparison tasks.

To speed up the EMD and Wasserstein distance computation, Cuturi (2013) proposed the Sinkhorn algorithm, which solves the entropic regularized optimization problem and computes the Wasserstein distance in quadratic order (i.e., $O(n^2)$). Kusner et al. (2015) proposed a relaxed Wasserstein Mover's distance (RWMD), which only considers the sum-to-one constraint. Because the RWMD computation is simple, the RWMD can be efficiently computed. Atasu & Mittelholzer (2019) proposed the approximate constrained transfer method, which iteratively adds constraints to the RWMD and improves its performance while maintaining the computational speed. To solve large-scale OT problems, Genevay et al. (2016) formulated the optimal transport problem as a concave maximization problem using a c-transform, and then solved the problem with the stochastic gradient descent (SGD) and the stochastic average gradient (SAG). Mensch & Peyré (2020) proposed the online Sinkhorn algorithm, which solves OT problems using a stream of samples.

Another approach is based on the sliced Wasserstein distance (SWD) (Rabin et al., 2011; Kolouri et al., 2016), which solves the optimal transport problem in a projected one-dimensional subspace. Because the one-dimensional optimal transport problem can be solved using sorting, the SWD can be computed in $O(n \log n)$. Several extensions of the SWD have been proposed. The generalized sliced Wasserstein distance extends the 1D sliced Wasserstein distance for multidimensional cases (Kolouri et al., 2019). The max-sliced Wasserstein distance determines the 1D subspace that maximizes the discrepancy between two distributions and then computes the optimal transport in the subspace (Mueller & Jaakkola, 2015; Deshpande et al., 2019). The subspace robust Wasserstein distance (SRWD) is a general extension of the max-sliced Wasserstein distance for multi-dimensional subspaces (Paty & Cuturi, 2019; Lin et al., 2020).

The tree-Wasserstein distance (TWD) (Evans & Matsen, 2012; Le et al., 2019) uses tree-based embedding, whereas the SWD uses 1D embedding. Because a chain of trees can represent 1D embedding, the TWD includes the SWD as a special case. Le et al. (2019) reported that the TWD can empirically outperform the SWD. The TWD has also been studied in the theoretical computer science community and can be computed using the QuadTree algorithm (Indyk & Thaper, 2003). Extensions of the TWD have also been studied; these include the unbalanced TWD (Sato et al., 2020; Le & Nguyen, 2021), supervised Wasserstein training (Takezawa et al., 2021), and tree barycenters (Takezawa et al., 2022). These approaches mainly focus on approximating the 1-Wasserstein distance with tree construction, where the weights of the edges are set by a constant number. Recently, Backurs et al. (2020) proposed a FlowTree that combines the QuadTree method and the cost matrix computed from vectors. They then showed that the FlowTree outperformed QuadTree-based approaches. Moreover, they theoretically guarantee that the QuadTree and FlowTree methods can approximate the nearest neighbor using the 1-Wasserstein distance. Dey & Zhang (2022) proposed an L1-embedding approach for approximating the 1-Wasserstein distance for the persistence diagram. However, there are no learning-based approaches for approximating the 1-Wasserstein distance in a

general setup. Thus, we focused on estimating the weight parameter of the TWD from data to approximate the 1-Wasserstein distance.

**Applications of the Optimal transport:** The optimal transport has been applied to various types of tasks including barycenter estimation (Cuturi & Doucet, 2014; Benamou et al., 2015), generative modeling (Arjovsky et al., 2017; Gulrajani et al., 2017; Kolouri et al., 2018), domain adaptation (Courty et al., 2017; Shen et al., 2018), inverse optimal transport (Li et al., 2019), differentiable sorting (Cuturi et al., 2019; Blondel et al., 2020; Xie et al., 2020), and modeling of population dynamics (Bunne et al., 2022). Moreover, using the Sinkhorn algorithm, we can solve the Wasserstein distance minimization problem in an end-to-end manner (Genevay et al., 2018). In this research direction, solving the Wasserstein distance minimization problem in the mini-batch setup is an important research topic (Nguyen et al., 2022).

Many OT-based applications were formulated as one of two minimization problems of the Wasserstein distance (Arjovsky et al., 2017), a minimization problem using the Sinkhorn algorithm (Genevay et al., 2018) or a minimization problem with 1D slicing (Kolouri et al., 2018). The tree-Wasserstein distance, which includes a sliced Wasserstein distance as a special case, has been applied to document classification (Le et al., 2019), topological data analysis (TDA) (Le et al., 2019), neural architecture search (NAS) (Nguyen et al., 2021), and barycenter estimation (Takezawa et al., 2022). Moreover, the TWD is known as the UniFrac distance (Lozupone & Knight, 2005), which is a phylogenetic method for comparing microbial communities. Thus, our proposed weighting technique for the TWD can be used for microbial community comparison.

## 3 Preliminary

In this section, we introduce the optimal transport, Wasserstein distances, and tree-Wasserstein distances.

### 3.1 Optimal transport and Wasserstein distance

**Optimal transport:** We computed the distance between two datasets $\{(\boldsymbol{x}_i, a_i)\}_{i=1}^n$ and $\{(\boldsymbol{x}_i', b_i)\}_{j=1}^{n'}$, where $\boldsymbol{x} \in \mathcal{X} \subset \mathbb{R}^d$, $\boldsymbol{x}' \in \mathcal{X}' \subset \mathbb{R}^d$, and $\sum_{i=1}^n a_i = 1$ and $\sum_{j=1}^{n'} b_j = 1$ are the probability masses. For example, in the text similarity setup, $\boldsymbol{x}$ and $\boldsymbol{x}'$ are precomputed word-embedding vectors, and $a_i$ is the frequency of word $\boldsymbol{x}_i$ in a document. Let $\mathcal{D} = \{\boldsymbol{x}_i\}_{i=1}^{N'}$ denote the entire set of vectors where $N'$ denotes the number of vectors. This study aims to measure the similarity between the two datasets: $\{(\boldsymbol{x}_i, a_i)\}_{i=1}^n$ and $\{(\boldsymbol{x}_i', b_i)\}_{j=1}^{n'}$.

In the Kantorovich relaxation of the OT, admissible couplings are defined by the set of transport plans between two discrete measures: $\mu = \sum_{i=1}^n a_i \delta_{\boldsymbol{x}_i}$ and $\nu = \sum_{j=1}^{n'} b_j \delta_{\boldsymbol{x}_j'}$.

$$U(\mu, \nu) = \{\boldsymbol{\Pi} \in \mathbb{R}_+^{n \times n'} : \boldsymbol{\Pi} \mathbf{1}_{n'} = \boldsymbol{a}, \boldsymbol{\Pi}^\top \mathbf{1}_n = \boldsymbol{b}\}, \tag{1}$$

where $\delta_{\boldsymbol{x}_i}$ is the Dirac at position $\boldsymbol{x}_i$, $\mathbf{1}_n$ is an $n$-dimensional vector whose elemets are 1, and $\boldsymbol{a} = (a_1, a_2, \ldots, a_n)^\top \in \mathbb{R}_+^n$ and $\boldsymbol{b} = (b_1, b_2, \ldots, b_{n'})^\top \in \mathbb{R}_+^{n'}$ are the probability masses of distributions $\mu$ and $\nu$, respectively.

Then, the OT problem between two discrete measures, $\mu$ and $\nu$, is given as

$$\min_{\boldsymbol{\Pi} \in U(\mu, \nu)} \sum_{i=1}^n \sum_{j=1}^{n'} \pi_{ij} c(\boldsymbol{x}_i, \boldsymbol{x}_j'), \tag{2}$$

where $\pi_{ij}$ is the $i,j$-th element of $\boldsymbol{\Pi}$, and $c(\boldsymbol{x}, \boldsymbol{x}')$ is a cost function (e.g., $c(\boldsymbol{x}, \boldsymbol{x}') = \|\boldsymbol{x} - \boldsymbol{x}'\|_2^2$). Eq. (2) is equivalent to the earth mover's distance, and it can be solved by using linear programming with $O(n^3)$ computation ($n = n'$). To accelerate the optimal transport computation, Cuturi (2013) proposed an entropic regularized optimal transport problem:

$$\min_{\boldsymbol{\Pi} \in U(\mu, \nu)} \sum_{i=1}^n \sum_{j=1}^{n'} \pi_{ij} c(\boldsymbol{x}_i, \boldsymbol{x}_j') - \epsilon H(\boldsymbol{\Pi}), \tag{3}$$

where $\epsilon \geq 0$ is the regularization parameter and $H(\mathbf{\Pi}) = -\sum_{i=1}^{n} \sum_{j=1}^{n'} \pi_{ij}(\log(\pi_{ij}) - 1)$ is the entropic regularization. The entropic regularized OT is equivalent to the EMD when $\epsilon = 0$. The entropic regularization problem can be efficiently solved using the Sinkhorn algorithm with $O(nn')$ cost. Moreover, the Sinkhorn algorithm can be efficiently implemented on a GPU, leading to a super-efficient matrix computation. Hence, the Sinkhorn algorithm is one of the most widely used OT solvers in ML.

**Wasserstein distance:** If we use distance function $d(\boldsymbol{x}, \boldsymbol{x}')$ for the cost $c(\boldsymbol{x}, \boldsymbol{x}')$ and $p \geq 1$, the $p$-Wasserstein distance of two discrete measures between two probability measures $\mu$ and $\nu$ is defined as

$$W_p(\mu, \nu) = \left( \inf_{\pi \in \mathcal{U}(\mu, \nu)} \int_{\mathcal{X} \times \mathcal{X}'} d(\boldsymbol{x}, \boldsymbol{x}') \mathrm{d}\pi(\boldsymbol{x}, \boldsymbol{x}')^p \right)^{1/p}, \tag{4}$$

where $\mathcal{U}$ denotes the set of joint probability distributions (Peyré et al., 2019) (Remark 2.13).

$$\mathcal{U}(\mu, \nu) = \{\pi \in \mathcal{M}_+^1(\mathcal{X} \times \mathcal{X}') : P_{\mathcal{X}\sharp}\pi = \mu, P_{\mathcal{X}'\sharp}\pi = \nu\}, \tag{5}$$

and $\mathcal{M}_+^1(\mathcal{X})$ is the set of probability measures, $P_{\mathcal{X}\sharp}$ and $P_{\mathcal{X}'\sharp}$ are push-forward operators (see Remark 2.2 in (Peyré et al., 2019)). The $p$-Wasserstein distance of the two discrete measures $\mu = \sum_{i=1}^{n} a_i \delta_{\boldsymbol{x}_i}$ and $\nu = \sum_{j=1}^{n'} b_j \delta_{\boldsymbol{x}_j'}$ is defined as:

$$W_p(\mu, \nu) = \left( \min_{\mathbf{\Pi} \in U(\mu, \nu)} \sum_{i=1}^{n} \sum_{j=1}^{n'} \pi_{ij} d(\boldsymbol{x}_i, \boldsymbol{x}_j')^p \right)^{1/p}. \tag{6}$$

From the definition of the $p$-Wasserstein distance, the 1-Wasserstein distance of the two discrete measures is given as

$$W_1(\mu, \nu) = \min_{\mathbf{\Pi} \in U(\mu, \nu)} \sum_{i=1}^{n} \sum_{j=1}^{n'} \pi_{ij} d(\boldsymbol{x}_i, \boldsymbol{x}_j'), \tag{7}$$

which is equivalent to Eq. (2) when $c(\boldsymbol{x}, \boldsymbol{x}') = d(\boldsymbol{x}, \boldsymbol{x}')$. For the distance function, the Euclidean distance $d(\boldsymbol{x}, \boldsymbol{x}') = \|\boldsymbol{x} - \boldsymbol{x}'\|_2$ and Manhattan distance $d(\boldsymbol{x}, \boldsymbol{x}') = \|\boldsymbol{x} - \boldsymbol{x}'\|_1$ are common choices. Note that Cuturi (2013) proposed the entropic regularized Wasserstein distance, although the entropic regularized formulation does not admit the triangle inequality; therefore, it is not a distance. Sinkhorn divergence was proposed to satisfy the metric axioms of the regularized entropic formulation (Genevay et al., 2018).

## 3.2 Tree-Wasserstein distance (TWD)

Although the Sinkhorn algorithm has a quadratic time complexity, it is still prohibitive when dealing with large vocabulary sets. The TWD has been attracting attention owing to its light computation and is thus used to approximate the 1-Wasserstein distance (Indyk & Thaper, 2003; Evans & Matsen, 2012; Le et al., 2019).

We define $\mathcal{T} = (V, E)$, where $V$ and $E$ are sets of nodes and edges, respectively. In particular, for a given tree $\mathcal{T}$, the EMD with a tree metric $d_{\mathcal{T}}(\boldsymbol{x}, \boldsymbol{x}')$ is called the TWD. The TWD admits the following closed-form expression:

$$W_{\mathcal{T}}(\mu, \nu) = \sum_{e \in E} w_e |\mu(\Gamma(v_e)) - \nu(\Gamma(v_e))|, \tag{8}$$

where $e$ is an edge index, $w_e \in \mathbb{R}_+$ is the edge weight of edge $e$, $v_e$ is the $e$th node index, and $\mu(\Gamma(v_e))$ is the total mass of the subtree with root $v_e$. The restriction on a tree metric is advantageous in terms of computational complexity; the TWD can be computed as $O(N)$, where $N$ is the number of nodes in a tree. It has been reported that the TWD compares favorably with the vanilla Wasserstein distance and is computationally efficient. Note that the sliced Wasserstein distance can be regarded as a special case of the TWD (Takezawa et al., 2022).

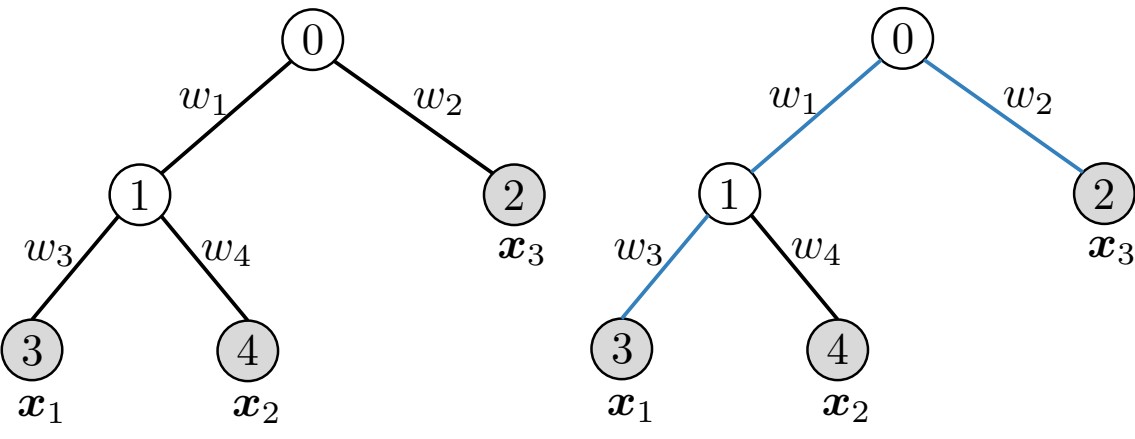

(a) Illustration of tree embedding.  (b) Shortest path between $\boldsymbol{x}_1$ and $\boldsymbol{x}_3$.

Figure 1: Illustration of tree embedding and the shortest path between $\boldsymbol{x}_1$ and $\boldsymbol{x}_3$ of the tree. All the input vectors are associated with the leaf nodes. The $\boldsymbol{b}_1$, $\boldsymbol{b}_2$, and $\boldsymbol{b}_3$ of the tree are given as $\boldsymbol{b}_1 = (1, 1, 0, 1, 0)^\top$, $\boldsymbol{b}_2 = (1, 1, 0, 0, 1)^\top$, and $\boldsymbol{b}_3 = (1, 0, 1, 0, 0)^\top$, respectively. In this case, $d_{\mathcal{T}}(\boldsymbol{x}_1, \boldsymbol{x}_3) = w_3 + w_1 + w_2$.

The TWD can be represented in matrix form as (Takezawa et al., 2021)

$$W_{\mathcal{T}}(\mu, \nu) = \|\mathrm{diag}(\boldsymbol{w})\boldsymbol{B}(\boldsymbol{a} - \boldsymbol{b})\|_1, \tag{9}$$

where $\boldsymbol{w} \in \mathbb{R}_+^N$ is the weight vector, $\boldsymbol{B} \in \{0, 1\}^{N \times N_{\mathrm{leaf}}}$ is a tree parameter, $[\boldsymbol{B}]_{i,j} = 1$ if node $i$ is the ancestor node of leaf node $j$ and zero otherwise, $N$ is the total number of nodes of a tree, $N_{\mathrm{leaf}}$ is the number of leaf nodes, and $\mathrm{diag}(\boldsymbol{w}) \in \mathbb{R}_+^{N \times N}$ is a diagonal matrix whose diagonal elements are $\boldsymbol{w}$. Figure 1a shows an example of tree embedding. In this tree, the total number of nodes is $N = 5$ and the number of leaf nodes is $N_{\mathrm{leaf}} = 3$. Note that $w_0$ is the weight of the root node. Because $\boldsymbol{a}$ and $\boldsymbol{b}$ are probability vectors (i.e., $\boldsymbol{a}^\top \mathbf{1} = 1$) and the elements of the first row of $\boldsymbol{B}$ are all 1, the weight of the root node is ignored in $W_{\mathcal{T}}(\mu, \nu)$.

To compute the TWD, the choice of the tree structure and weight is crucial for approximating the Wasserstein distance. Indyk & Thaper (2003) proposed the QuadTree algorithm, which constructs a tree by recursively splitting a space into four regions (quad) and setting the weight parameter as $2^{-\ell(e)}$ ($\ell(e)$ is the depth of an edge $e$). For the QuadTree, the TWD is defined as

$$W_{\mathcal{T}}(\mu, \nu) = \sum_{e \in E} 2^{-\ell(e)} |\mu(\Gamma(v_e)) - \nu(\Gamma(v_e))|. \tag{10}$$

Le et al. (2019) proposed a clustering-based tree construction. More recently, Takezawa et al. (2021) proposed training a tree using continuous optimization in a supervised learning setup.

Previous research related to the TWD has focused on tree construction. However, learning edge weights in the TWD have not been thoroughly examined. Thus, in this study, we propose a weight estimation procedure for the TWD to approximate the Wasserstein distance.

## 4 Proposed method

In this section, we propose a tree weight estimation method using non-negative Lasso to accurately approximate the 1-Wasserstein distance.

### 4.1 Tree weight estimation using Lasso

Because the TWD is a 1-Wasserstein distance with a tree metric, it is natural to estimate the weight that can approximate it with an arbitrary distance $d(\boldsymbol{x}, \boldsymbol{x}')$. Thus, we propose estimating the weight $\boldsymbol{w}$ by minimizing the error between the 1-Wasserstein distance and the distance $d(\boldsymbol{x}, \boldsymbol{x}')$ and TWD. First, we show that the TWD can approximate the 1-Wasserstein distance arbitrarily with some trees.

---

**Algorithm 1** Sliced weight estimation with trees

---

1: **Input:** The matrix $\boldsymbol{X}$, the regularization parameter $\lambda_1 \geq 0$, and a set of indices $\Omega$.
2: **for** $t = 1, 2, \ldots, T$ **do**
3:     random.seed($i$)
4:     $\boldsymbol{B}_t := \text{QuadTree}(\boldsymbol{X})$ or $\boldsymbol{B}_t := \text{ClusterTree}(\boldsymbol{X})$
5:     Compute $\boldsymbol{z}_{i,j}^{(t)}$ from $\boldsymbol{B}_t$ and $d(\boldsymbol{x}_i, \boldsymbol{x}_j), (i,j) \in \Omega$
6:     $\widehat{\boldsymbol{w}}_t := \text{argmin}_{\boldsymbol{w} \in \mathbb{R}_+^{N_t}} \sum_{(i,j) \in \Omega} (d(\boldsymbol{x}_i, \boldsymbol{x}_j) - \boldsymbol{w}^\top \boldsymbol{z}_{i,j}^{(t)})^2 + \lambda \|\boldsymbol{w}\|_1$.
7: **end for**
8: **return** $\{(\boldsymbol{B}_t, \widehat{\boldsymbol{w}}_t)\}_{t=1}^T$

---

**Proposition 1** *For any measure $\mu = \sum_{i=1}^n a_i \delta_{\boldsymbol{x}_i}$ and $\nu = \sum_{j=1}^{n'} b_j \delta_{\boldsymbol{x}'_j}$ and the distance function $d \colon \mathcal{X} \times \mathcal{X} \to \mathbb{R}$, there exists a tree $\mathcal{T}$ such that $W_1(\mu, \nu) = W_{\mathcal{T}}(\mu, \nu)$ and $\boldsymbol{\Pi}^*(\mu, \nu) = \boldsymbol{\Pi}_{\mathcal{T}}^*(\mu, \nu)$ holds, where $\boldsymbol{\Pi}^*(\mu, \nu)$ is an optimal solution to Eq. (2) with cost $c = d$, and $\boldsymbol{\Pi}_{\mathcal{T}}^*(\mu, \nu)$ is an optimal solution with the cost function being the shortest path distance $d_{\mathcal{T}}$ on tree $\mathcal{T}$.*

The proof is provided in section A. This proposition indicates that the TWD has sufficient expressive power to model the 1–Wasserstein distance.

**Proposition 2** *(Le et al., 2019)*

$$W_{\mathcal{T}}(\mu, \nu) = \inf_{\pi \in \mathcal{U}(\mu, \nu)} \int_{\mathcal{X} \times \mathcal{X}'} d_{\mathcal{T}}(\boldsymbol{x}, \boldsymbol{x}') \mathrm{d}\pi(\boldsymbol{x}, \boldsymbol{x}'), \tag{11}$$

*where $d_{\mathcal{T}}(\boldsymbol{x}, \boldsymbol{x}')$ is the length of the (unique) path between $\boldsymbol{x}$ and $\boldsymbol{x}'$ on a tree (i.e., the shortest path on the tree), and $\mathcal{U}$ is a set of joint probability distributions.*

This proposition indicates that the 1-Wasserstein distance with an arbitrary distance metric and the TWD are equivalent (i.e., $W_1(\mu, \nu) = W_{\mathcal{T}}(\mu, \nu)$) if $d(\boldsymbol{x}, \boldsymbol{x}') = d_{\mathcal{T}}(\boldsymbol{x}, \boldsymbol{x}')$. Thus, we aim to fit $d(\boldsymbol{x}, \boldsymbol{x}')$ using $d_{\mathcal{T}}(\boldsymbol{x}, \boldsymbol{x}')$, where the tree structure and edge weights are the potential learning parameters. Figure 1b illustrates the shortest distance between $\boldsymbol{x}_1$ and $\boldsymbol{x}_3$. In this case, $d_{\mathcal{T}}(\boldsymbol{x}_1, \boldsymbol{x}_3) = \sum_{k \in \text{Path}(\boldsymbol{x}_1, \boldsymbol{x}_3)} w_k = w_3 + w_1 + w_2$; thus, a closed-form expression for the shortest path distance can be obtained as follows:

**Proposition 3** *We denote $\boldsymbol{B} \in \{0,1\}^{N \times N_{leaf}} = [\boldsymbol{b}_1, \boldsymbol{b}_2, \ldots, \boldsymbol{b}_{N_{leaf}}]$ and $\boldsymbol{b}_i \in \{0,1\}^N$. The shortest path distance between leaves $i$ and $j$ can be represented as*

$$d_{\mathcal{T}}(\boldsymbol{x}_i, \boldsymbol{x}_j) = \boldsymbol{w}^\top (\boldsymbol{b}_i + \boldsymbol{b}_j - 2\boldsymbol{b}_i \circ \boldsymbol{b}_j). \tag{12}$$

*(Proof)* *The shortest distance between leaves $i$ and $j$ can be written as*

$$
\begin{aligned}
d_{\mathcal{T}}(\boldsymbol{x}_i, \boldsymbol{x}_j) &= \sum_{k \in \text{Path}(\boldsymbol{x}_i, \boldsymbol{x}_j)} w_k \\
&= \sum_{k \in \mathcal{S}_i} w_k + \sum_{k' \in \mathcal{S}_j} w_{k'} - 2 \sum_{k \in \mathcal{S}_i \cap \mathcal{S}_j} w_k \\
&= \boldsymbol{w}^\top \boldsymbol{b}_i + \boldsymbol{w}^\top \boldsymbol{b}_j - 2\boldsymbol{w}^\top (\boldsymbol{b}_i \circ \boldsymbol{b}_j) \\
&= \boldsymbol{w}^\top (\boldsymbol{b}_i + \boldsymbol{b}_j - 2\boldsymbol{b}_i \circ \boldsymbol{b}_j),
\end{aligned}
$$

*where $\mathcal{S}_i$ includes the set of ancestor node indices of leaf $i$ and node index of leaf $i$.* □

Given the closed-form expression in Proposition 3, we formulated the edge-weight estimation problem as follows. First, we assumed that the tree was obtained using a tree construction algorithm, such as the

QuadTree (Indyk & Thaper, 2003) and ClusterTree (Le et al., 2019) algorithms, and we fixed the tree structure (i.e., fixed $\boldsymbol{B}$). We then propose the following optimization for learning $\boldsymbol{w}$:

$$\widehat{\boldsymbol{w}} := \underset{\boldsymbol{w} \in \mathbb{R}_+^N}{\operatorname{argmin}} \sum_{i,j=1}^{m} (d(\boldsymbol{x}_i, \boldsymbol{x}_j) - \boldsymbol{w}^\top \boldsymbol{z}_{i,j})^2 + \lambda \|\boldsymbol{w}\|_1, \tag{13}$$

where $\boldsymbol{z}_{i,j} = \boldsymbol{b}_i + \boldsymbol{b}_j - 2\boldsymbol{b}_i \circ \boldsymbol{b}_j \in \mathbb{R}_+^N$, $\|\boldsymbol{w}\|_1$ is the L1-norm, and $\lambda \geq 0$ is a regularization parameter. This optimization problem is convex with respect to $\boldsymbol{w}$. We can easily solve the problem using an off-the-shelf non-negative Lasso solver. $\boldsymbol{z}_{i,j}$ is a sparse vector with only $2H \ll N$ elements, where $H$ is the tree depth. Thus, we can store all of $\boldsymbol{z}$ in a sparse format and efficiently solve the Lasso problem. Moreover, owing to the L1-regularization, we can make most of the weights zero, which corresponds to merging the nodes of the tree. Thus, with this optimization, we are not only able to estimate the weight of the tree but also compress its size.

We also propose a subsampled variant of the regression method.

$$\widehat{\boldsymbol{w}} := \underset{\boldsymbol{w} \in \mathbb{R}_+^N}{\operatorname{argmin}} \sum_{(i,j) \in \Omega} (d(\boldsymbol{x}_i, \boldsymbol{x}_j) - \boldsymbol{w}^\top \boldsymbol{z}_{i,j})^2 + \lambda \|\boldsymbol{w}\|_1, \tag{14}$$

where $\Omega$ denotes a set of indices. In this study, we randomly sub-sampled indices.

Although our proposed method is specifically designed for the 1-Wasserstein distance approximation, it can also be used to approximate any distance by using the shortest path distance on a tree.

## 4.2 Tree-slice extension

Thus far, we have focused on estimating distance using a single tree. However, the approximation performance is highly dependent on the tree that is fixed before weight estimation. To address this issue, we can use the tree-sliced Wasserstein distance (Le et al., 2019)

$$\bar{W}_\mathcal{T}(\mu, \nu) = \frac{1}{T} \sum_{t=1}^{T} W_{\mathcal{T}_t}(\mu, \nu), \tag{15}$$

where $W_{\mathcal{T}_t}(\mu, \nu)$ is the TWD with the $t$-th generated tree and $T$ is the total number of trees. In previous studies, the TWD outperformed approaches based on a single tree. The matrix version of the tree-sliced Wasserstein distance can be written as

$$\bar{W}_\mathcal{T}(\mu, \nu) = \frac{1}{T} \sum_{t=1}^{T} \|\operatorname{diag}(\boldsymbol{w}_t) \boldsymbol{B}_t (\boldsymbol{a} - \boldsymbol{b})\|_1, \tag{16}$$

where $\boldsymbol{w}_t$ and $\boldsymbol{B}_t$ are the weight and parameters of the $t$-th tree, respectively.

In this study, we extended the single-tree weight estimation to multiple trees. To this end, we separately estimated $\widehat{\boldsymbol{w}}_t$ from each tree and then averaged the estimates.

$$\bar{W}_\mathcal{T}(\mu, \nu) = \frac{1}{T} \sum_{t=1}^{T} \|\operatorname{diag}(\widehat{\boldsymbol{w}}_t) \boldsymbol{B}_t (\boldsymbol{a} - \boldsymbol{b})\|_1, \tag{17}$$

where

$$\widehat{\boldsymbol{w}}_t := \underset{\boldsymbol{w} \in \mathbb{R}_+^{N_t}}{\operatorname{argmin}} \sum_{(i,j) \in \Omega} (d(\boldsymbol{x}_i, \boldsymbol{x}_j) - \boldsymbol{w}^\top \boldsymbol{z}_{i,j}^{(t)})^2 + \lambda \|\boldsymbol{w}\|_1, \tag{18}$$

$\boldsymbol{z}_{i,j}^{(t)} \in \mathbb{R}_+^{N_t}$ is computed from $\boldsymbol{B}_t$, where $N_t$ is the number of nodes in the $t$th tree. We call the proposed methods with the QuadTree and ClusterTree algorithms, the sliced qTWD and sliced-cTWD, respectively. Algorithm 1 is the pseudocode for the weight estimation procedure that uses trees. If we set $T = 1$, this corresponds to the single-tree case.

Table 1: Results of the 1-Wasserstein distance approximation for the Twitter dataset. We report the averaged mean absolute error (MAE), averaged Pearson's correlation coefficients (PCC), and averaged number of nodes used for computation, respectively. For the tree-sliced variants, we empirically fixed the number of sliced as $T = 3$. For the MAE and PCC computation, we computed the MAE and PCC between the 1-Wasserstein distance computed by EMD with Euclidean distance and tree counterparts, and we ran the algorithm 10 times by changing the random seed.

| | Twitter | | |
| Methods | MAE | PCC | Nodes |
|---|---|---|---|
| QuadTree | $0.486 \pm 0.060$ | $0.702 \pm 0.077$ | $5097.2 \pm 56.3$ |
| qTWD ($\lambda = 10^{-3}$) | $0.053 \pm 0.003$ | $0.880 \pm 0.031$ | $4976.4 \pm 103.6$ |
| qTWD ($\lambda = 10^{-2}$) | $0.053 \pm 0.003$ | $0.879 \pm 0.030$ | $4542.2 \pm 56.9$ |
| qTWD ($\lambda = 10^{-1}$) | $0.051 \pm 0.002$ | $0.873 \pm 0.030$ | $4328.4 \pm 42.0$ |
| ClusterTree | $8.028 \pm 0.262$ | $0.674 \pm 0.056$ | $5717.5 \pm 47.1$ |
| cTWD ($\lambda = 10^{-3}$) | $0.039 \pm 0.002$ | $0.885 \pm 0.025$ | $5674.1 \pm 50.0$ |
| cTWD ($\lambda = 10^{-2}$) | $0.038 \pm 0.002$ | $0.881 \pm 0.026$ | $5481.0 \pm 31.6$ |
| cTWD ($\lambda = 10^{-1}$) | $0.036 \pm 0.002$ | $0.867 \pm 0.029$ | $4195.2 \pm 63.1$ |
| Sliced-QuadTree | $0.463 \pm 0.045$ | $0.796 \pm 0.048$ | $15164.6 \pm 141.4$ |
| Sliced-qTWD ($\lambda = 10^{-3}$) | $0.053 \pm 0.003$ | $0.890 \pm 0.032$ | $14702.7 \pm 256.5$ |
| Sliced-qTWD ($\lambda = 10^{-2}$) | $0.052 \pm 0.002$ | $0.892 \pm 0.031$ | $13581.1 \pm 82.8$ |
| Sliced-qTWD ($\lambda = 10^{-1}$) | $0.050 \pm 0.002$ | $0.892 \pm 0.028$ | $13026.3 \pm 82.6$ |
| Sliced-ClusterTree | $7.989 \pm 0.139$ | $0.804 \pm 0.035$ | $17129.5 \pm 58.9$ |
| Sliced-cTWD ($\lambda = 10^{-3}$) | $0.035 \pm 0.002$ | $0.929 \pm 0.017$ | $17002.6 \pm 61.9$ |
| Sliced-cTWD ($\lambda = 10^{-2}$) | $0.034 \pm 0.001$ | $0.929 \pm 0.017$ | $16414.7 \pm 61.8$ |
| Sliced-cTWD ($\lambda = 10^{-1}$) | $0.029 \pm 0.001$ | $0.930 \pm 0.019$ | $12565.3 \pm 104.0$ |

### 4.3 Computational cost

The computational cost of the proposed algorithm consists of three parts: computing a tree, estimating the weight, and computing the TWD. In many Wasserstein distance applications (e.g., document retrieval), it is necessary to compute the tree and weight only once. In this study, we employed the QuadTree and ClusterTree algorithms, which are efficient methods for tree construction. The computational complexity of the QuadTree is linear with respect to the number of samples $O(dN_{\text{leaf}} \log 1/\epsilon)$, where $d$ is the number of input dimensions and $\epsilon > 0$ is the accuracy. For the ClusterTree (Le et al., 2019), we employed farthest-point clustering (Gonzalez, 1985), and the computational complexity of $O(dHN_{\text{leaf}} \log K)$, where $H$ is the depth of the tree and $K$ is the number of clusters. For weight estimation, we employed the fast iterative shrinkage thresholding algorithm (FISTA) (Beck & Teboulle, 2009), which is the ISTA with Nesterov's accelerated gradient descent (Nesterov, 1983). FISTA converges to the optimal value as $O(1/k^2)$, where $k$ is the number of the iteration. For example, the dominant computation of FISTA on the Lasso problem $\|\boldsymbol{y} - \boldsymbol{A}\boldsymbol{w}\|_2^2 + \lambda\|\boldsymbol{w}\|$ is the matrix-vector multiplication of $\boldsymbol{A}$ and $\boldsymbol{A}^\top$. In our case, the size of $\boldsymbol{A}$ is $|\Omega| \times N$, which includes $|\Omega|N$ multiplications and summations for each iteration. In practice, the computational cost of matrix-vector multiplication is small (see Table 5). Overall, when the number of vectors $N_{\text{leaf}}$ is $10{,}000$ and $30{,}000$, tree construction and weight estimation can be performed efficiently with a single CPU. For inference, the TWD can be efficiently computed using our tree-based approach because its computational complexity is $O(N)$, where $N$ is the total number of nodes of a tree. For the tree-sliced variant case, the computational cost of the tree-sliced Wasserstein is $T$ times larger than that of a single-tree case.

## 5 Experiments

We evaluated our proposed methods using the Twitter, BBCSport, and Amazon datasets.

Table 2: Results of the 1-Wasserstein distance approximation for the BBCSports dataset. We report the averaged mean absolute error (MAE), averaged Pearson's correlation coefficients (PCC), and averaged number of nodes used for computation, respectively. For tree-sliced variants, we empirically fixed the number of sliced as $T = 3$. For the MAE and PCC computation, we computed the MAE and PCC between the 1-Wasserstein distance computed by EMD with Euclidean distance and the tree counterparts. Further, we ran the algorithm 10 times by changing the random seed.

| Methods | BBCSport | | |
| | MAE | PCC | Nodes |
|---|---|---|---|
| QuadTree | $0.512 \pm 0.023$ | $0.851 \pm 0.031$ | $11463.1 \pm 153.6$ |
| qTWD ($\lambda = 10^{-3}$) | $0.098 \pm 0.005$ | $0.923 \pm 0.016$ | $11016.4 \pm 21.6$ |
| qTWD ($\lambda = 10^{-2}$) | $0.095 \pm 0.007$ | $0.921 \pm 0.017$ | $9968.5 \pm 47.1$ |
| qTWD ($\lambda = 10^{-1}$) | $0.073 \pm 0.022$ | $0.878 \pm 0.046$ | $8393.8 \pm 304.9$ |
| cTree | $5.842 \pm 0.171$ | $0.799 \pm 0.041$ | $12217.7 \pm 57.4$ |
| cTWD ($\lambda = 10^{-3}$) | $0.028 \pm 0.005$ | $0.902 \pm 0.020$ | $12145.3 \pm 56.5$ |
| cTWD ($\lambda = 10^{-2}$) | $0.023 \pm 0.004$ | $0.894 \pm 0.022$ | $11031.8 \pm 99.2$ |
| cTWD ($\lambda = 10^{-1}$) | $0.023 \pm 0.003$ | $0.866 \pm 0.027$ | $6680.7 \pm 97.8$ |
| Sliced-QuadTree | $0.498 \pm 0.020$ | $0.901 \pm 0.022$ | $34205.7 \pm 284.6$ |
| Sliced-qTWD ($\lambda = 10^{-3}$) | $0.100 \pm 0.003$ | $0.945 \pm 0.010$ | $32447.4 \pm 586.0$ |
| Sliced-qTWD ($\lambda = 10^{-2}$) | $0.096 \pm 0.004$ | $0.945 \pm 0.011$ | $29966.5 \pm 75.0$ |
| Sliced-qTWD ($\lambda = 10^{-1}$) | $0.073 \pm 0.014$ | $0.925 \pm 0.020$ | $25647.7 \pm 649.4$ |
| Sliced-ClusterTree | $5.830 \pm 0.080$ | $0.863 \pm 0.026$ | $36644.2 \pm 75.1$ |
| Sliced-cTWD ($\lambda = 10^{-3}$) | $0.026 \pm 0.002$ | $0.948 \pm 0.009$ | $36405.2 \pm 80.5$ |
| Sliced-cTWD ($\lambda = 10^{-2}$) | $0.021 \pm 0.002$ | $0.943 \pm 0.009$ | $33024.7 \pm 112.9$ |
| Sliced-cTWD ($\lambda = 10^{-1}$) | $0.018 \pm 0.004$ | $0.924 \pm 0.013$ | $19947.2 \pm 148.3$ |

## 5.1 Setup

The approximation ability of the proposed method was evaluated. More specifically, we measured the error and correlation between the 1-Wasserstein distance using Euclidean norm and tree-based methods. For the tree-based methods, we evaluated QuadTree (Indyk & Thaper, 2003) and ClusterTree (Le et al., 2019). For the ClusterTree, we set the number of clusters $K = 5$ for all experiments. For the 1-Wasserstein distance, we used the Python optimal transport (POT) package [1]. For the tree methods, we first constructed a tree using an entire word embedding vector $\boldsymbol{X}$ and then computed the Wasserstein distance with the tree. For the proposed methods, we selected the regularization parameter from $\{10^{-3}, 10^{-2}, 10^{-1}\}$. The qTWD and cTWD are both proposed methods, and qTWD and cTWD employ a QuadTree and ClusterTree, respectively. We compared all methods using the Twitter, BBCSport, and Amazon datasets [2]. We further evaluated the group-sliced TWD using a QuadTree and ClusterTree. We randomly subsampled 100 document pairs and computed the mean absolute error (MAE) and Pearson's correlation coefficient (PCC) between the 1-Wasserstein distance computed by EMD with the Euclidean distance and its tree counterparts. Moreover, we reported the number of nonzero tree weights. In this experiment, we set the number of slices to $T = 3$ and the regularization parameters $\lambda = \{10^{-3}, 10^{-2}, 10^{-1}\}$. We used SPAMS to solve the Lasso problem [3]. For all methods, we ran the algorithm 10 times by changing the random seed. We evaluated all the methods using Xeon CPU E5-2690 v4 (2.60 GHz) and Xeon CPU E7-8890 v4 (2.20 GHz). Note that because the number of vectors $\boldsymbol{x}_i$ tends to be large for real-world applications, we evaluated only the subsampled training method.

---

[1] https://pythonot.github.io/index.html
[2] https://github.com/gaohuang/S-WMD
[3] http://thoth.inrialpes.fr/people/mairal/spams/

Table 3: Results of the 1-Wasserstein distance approximation for the Amazon dataset. We report the averaged mean absolute error (MAE), averaged Pearson's correlation coefficients (PCC), and averaged number of nodes used for computation. For the tree-sliced variants, we empirically fixed the number of sliced as $T = 3$. For the MAE and PCC computation, we computed the MAE and PCC between the 1-Wasserstein distance computed by EMD with Euclidean distance and the tree counterparts. Further, we ran the algorithm 10 times by changing the random seed.

| Methods | Amazon | | |
| | MAE | PCC | Nodes |
| --- | --- | --- | --- |
| QuadTree | $0.573 \pm 0.034$ | $0.596 \pm 0.070$ | $34203.0 \pm 672.6$ |
| qTWD ($\lambda = 10^{-3}$) | $0.072 \pm 0.002$ | $0.789 \pm 0.040$ | $32132.7 \pm 956.4$ |
| qTWD ($\lambda = 10^{-2}$) | $0.067 \pm 0.005$ | $0.783 \pm 0.035$ | $28443.3 \pm 312.7$ |
| qTWD ($\lambda = 10^{-1}$) | $0.051 \pm 0.009$ | $0.732 \pm 0.064$ | $16983.9 \pm 2202.4$ |
| ClusterTree | $8.074 \pm 0.073$ | $0.759 \pm 0.031$ | $33430.0 \pm 51.8$ |
| cTWD ($\lambda = 10^{-3}$) | $0.036 \pm 0.002$ | $0.785 \pm 0.046$ | $32642.3 \pm 59.7$ |
| cTWD ($\lambda = 10^{-2}$) | $0.033 \pm 0.002$ | $0.778 \pm 0.047$ | $28701.8 \pm 134.8$ |
| cTWD ($\lambda = 10^{-1}$) | $0.028 \pm 0.002$ | $0.768 \pm 0.047$ | $9879.8 \pm 347.0$ |
| Sliced-QuadTree | $0.552 \pm 0.026$ | $0.759 \pm 0.047$ | $102274.2 \pm 992.5$ |
| Sliced-qTWD ($\lambda = 10^{-3}$) | $0.072 \pm 0.002$ | $0.838 \pm 0.024$ | $94737.4 \pm 1756.5$ |
| Sliced-qTWD ($\lambda = 10^{-2}$) | $0.067 \pm 0.003$ | $0.849 \pm 0.023$ | $85604.9 \pm 514.1$ |
| Sliced-qTWD ($\lambda = 10^{-1}$) | $0.047 \pm 0.009$ | $0.838 \pm 0.039$ | $53689.9 \pm 4423.5$ |
| Sliced-ClusterTree | $8.071 \pm 0.066$ | $0.822 \pm 0.021$ | $100,251.3 \pm 89.6$ |
| Sliced-cTWD ($\lambda = 10^{-3}$) | $0.035 \pm 0.002$ | $0.888 \pm 0.020$ | $97891.5 \pm 89.8$ |
| Sliced-cTWD ($\lambda = 10^{-2}$) | $0.031 \pm 0.002$ | $0.884 \pm 0.019$ | $86068.8 \pm 202.1$ |
| Sliced-cTWD ($\lambda = 10^{-1}$) | $0.022 \pm 0.001$ | $0.870 \pm 0.012$ | $29540.0 \pm 519.6$ |

## 5.2 Results of the approximation of the 1-Wasserstein distance

Tables 1,2, and 3 present the experimental results for the Twitter, BBCSport, and Amazon datasets, respectively. Evidently, the qTWD and cTWD can obtain a small MAE and accurately approximate the original 1-Wasserstein distance. In contrast, the QuadTree and ClusterTree had larger MAE values than the proposed methods. Although the QuadTree has a theoretical guarantee, its MAE is one order of magnitude larger than that of the proposed method. Because ClusterTree is based on clustering, and the tree construction is independent of the scaling of vectors, it cannot guarantee a good approximation of the 1-Wasserstein distance. Our proposed method can approximate the 1-Wasserstein distance even if ClusterTree is used. For the PCC, the qTWD and cTWD outperformed the vanilla QuadTree and ClusterTree when we set a small $\lambda$. Moreover, even when using half of the nodes (i.e., $\lambda = 10^{-1}$), the PCCs of qTWD and cTWD were comparable to those of QuadTree and ClusterTree. Figure 2 shows scatter plots for each method. We can observe that all methods correlate with the 1-Wasserstein distance. However, it is evident that the proposed method has a scale similar to that of the original WD, whereas QuadTree and ClusterTree have larger values than the original WD. Figure 3 shows the effect of the training sample size used for training in Eq. (14). In these experiments, the number of samples was varied as $25,000$, $50,000$, $75,000$, and $100,000$. We found that the PCC could be improved by using more samples. In contrast, the MAE was small, even when we used a relatively small number of samples.

For the tree-sliced variants, as reported in (Le et al., 2019), the sliced QuadTree and sliced ClusterTree outperformed their non-sliced counterparts. Although there was no gain in MAE, our proposed methods, Sliced-qTWD and Sliced-cTWD, can significantly improve the PCC values. Furthermore, because we use the distance to retrieve similar documents, we can obtain similar results using the Wasserstein distance. More interestingly, for the Amazon dataset, the sliced-cTWD ($\lambda = 10^{-1}$) had a PCC value of 0.870 with 29,540 nodes, whereas the cTWD ($\lambda = 10^{-3}$) had a PCC value of 0.785 with 32,642.3 nodes. Thus, the

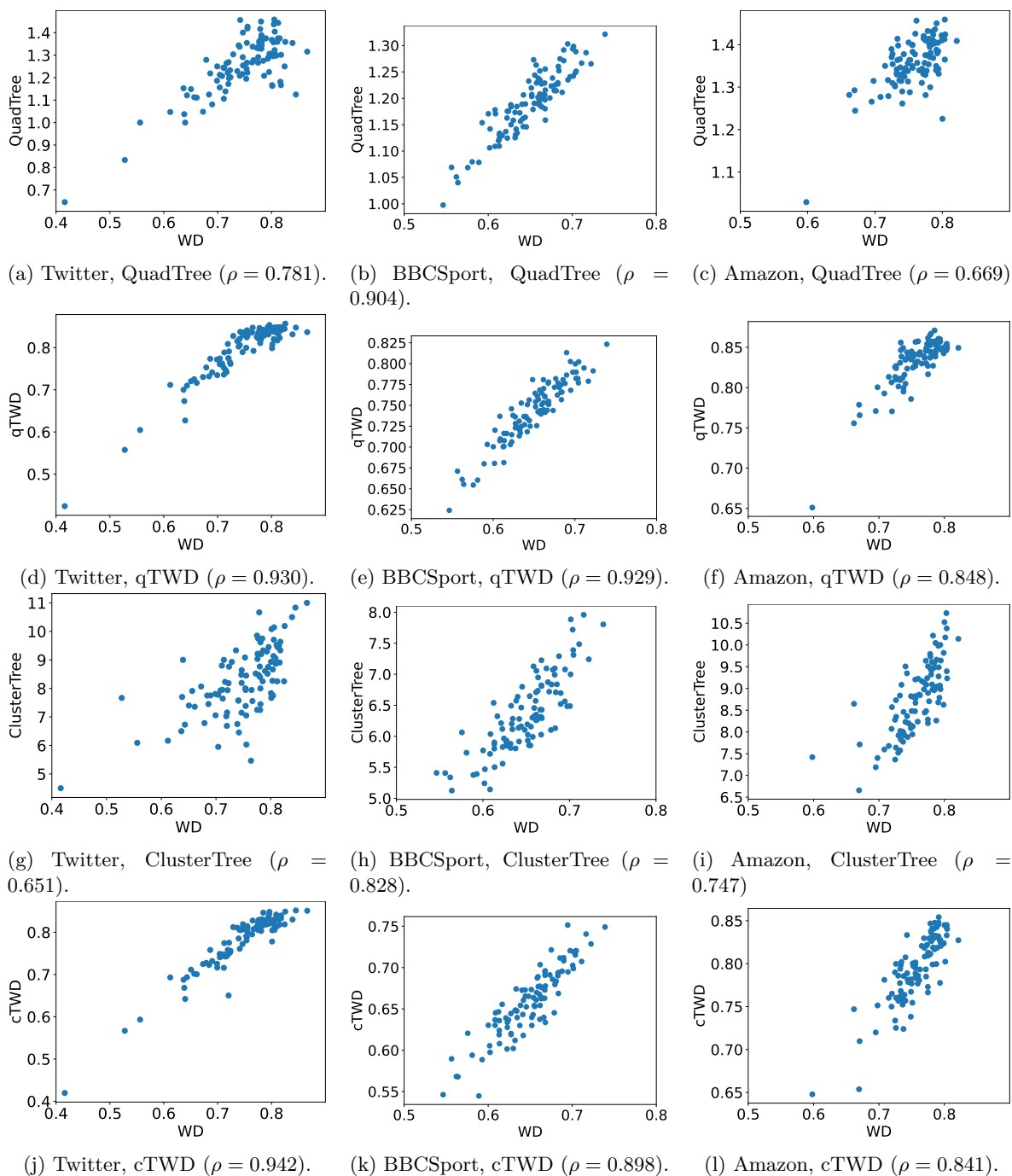

(a) Twitter, QuadTree ($\rho = 0.781$).

(b) BBCSport, QuadTree ($\rho = 0.904$).

(c) Amazon, QuadTree ($\rho = 0.669$)

(d) Twitter, qTWD ($\rho = 0.930$).

(e) BBCSport, qTWD ($\rho = 0.929$).

(f) Amazon, qTWD ($\rho = 0.848$).

(g) Twitter, ClusterTree ($\rho = 0.651$).

(h) BBCSport, ClusterTree ($\rho = 0.828$).

(i) Amazon, ClusterTree ($\rho = 0.747$)

(j) Twitter, cTWD ($\rho = 0.942$).

(k) BBCSport, cTWD ($\rho = 0.898$).

(l) Amazon, cTWD ($\rho = 0.841$).

Figure 2: Scatter plots of the Twitter, BBCSport, and Amazon datasets.

sliced-cTWD significantly outperformed cTWD. Thus, to obtain a low approximation error, one strategy is to use the sliced version and to prune unimportant nodes using Lasso.

## 5.3 Results of document classification

We evaluated the document classification experiment tasks with the 1-Wasserstein distance (Sinkhorn algorithm), QuadTree, ClusterTree, qTWD, cTWD, and their sliced counterparts. For the proposed method, we set the regularization parameter to $\lambda = 10^{-3}$. In this experiment, we used 90% of the samples for training

Figure 3: MAE and PCC with respect to the number of training samples. In this experiments, we computed the MAE and PCC for $m \in \{25,000, 50,000, 75,000, 100,000\}$. Note that the MAEs of the ClusterTree are larger than 1, and they do not appear in (d),(e), and (f).

and the remaining samples for the test. We then used the ten nearest neighbor classification methods. We ran the algorithm five times by changing the random seed and report the average mean accuracy.

Table 4: Document classification results for the Twitter, BBCSport, and Amazon datasets. We used the ten nearest neighbor classifiers. For the Sinkhorn algorithm, we set the regularization parameter with $10^{-2}$ and the maximum iteration with 100, respectively. We ran the algorithms five times by changing the random seed and report the averaged classification accuracy. For the proposed method, we set the regularization parameter as $\lambda = 10^{-3}$.

| Methods | Twitter | BBCSport | Amazon |
|---|---|---|---|
| WMD (Sinkhorn) | $0.675 \pm 0.033$ | $0.973 \pm 0.015$ | $0.903 \pm 0.006$ |
| QuadTree | $0.701 \pm 0.027$ | $0.970 \pm 0.016$ | $0.865 \pm 0.001$ |
| qTWD | $0.691 \pm 0.028$ | $0.967 \pm 0.014$ | $0.847 \pm 0.036$ |
| Sliced-QuadTree | $0.694 \pm 0.017$ | $0.970 \pm 0.020$ | $0.877 \pm 0.001$ |
| Sliced-qTWD | $0.697 \pm 0.027$ | $0.967 \pm 0.014$ | $0.887 \pm 0.011$ |
| ClusterTree | $0.683 \pm 0.019$ | $0.901 \pm 0.056$ | $0.873 \pm 0.010$ |
| cTWD | $0.699 \pm 0.032$ | $0.962 \pm 0.016$ | $0.878 \pm 0.006$ |
| Sliced-ClusterTree | $0.694 \pm 0.010$ | $0.929 \pm 0.037$ | $0.900 \pm 0.011$ |
| Sliced-cTWD | $0.700 \pm 0.021$ | $0.970 \pm 0.018$ | $0.905 \pm 0.010$ |

Table 5: Time comparisons for the Twitter, BBCSport, and Amazon datasets. For training, we computed the time using Xeon CPU E5-2690 v4 (2.60 GHz). For the test, we ran all the methods with an A6000 GPU. We ran the algorithms five times by changing the random seed and report the averaged computational time. We computed the average time consumption for comparing entire documents in the training set with one document for the test. The number in the parenthesis of each dataset is the number of the word vectors used in this study.

| Dataset | Methods | Training (CPU) | | | | Test (GPU) |
|---|---|---|---|---|---|---|
| | | Cost matrix | Tree const. | Training set gen. | Lasso | |
| | Sinkhorn | 3.0 | n/a | n/a | n/a | 0.159 |
| | QuadTree | n/a | 8.0 | n/a | n/a | $9.1 \times 10^{-5}$ |
| Twitter | ClusterTree | n/a | 6.0 | n/a | n/a | $8.8 \times 10^{-5}$ |
| ($N_{\text{leaf}} = 4,489$) | qTWD ($\lambda = 10^{-3}$) | n/a | 8.0 | 39.4 | 4.1 | $8.8 \times 10^{-5}$ |
| | cTWD ($\lambda = 10^{-3}$) | n/a | 6.0 | 49.8 | 11.7 | $9.0 \times 10^{-5}$ |
| | Sinkhorn | 14.7 | n/a | n/a | n/a | 0.125 |
| | QuadTree | n/a | 24.9 | n/a | n/a | $8.7 \times 10^{-5}$ |
| BBCsport | ClusterTree | n/a | 21.3 | n/a | n/a | $8.8 \times 10^{-5}$ |
| ($N_{\text{leaf}} = 10,103$) | qTWD ($\lambda = 10^{-3}$) | n/a | 24.9 | 95.6 | 4.5 | $8.7 \times 10^{-5}$ |
| | cTWD ($\lambda = 10^{-3}$) | n/a | 21.3 | 109.7 | 12.4 | $8.7 \times 10^{-5}$ |
| | Sinkhorn | 183.4 | n/a | n/a | n/a | 2.24 |
| | QuadTree | n/a | 162.2 | n/a | n/a | $1.1 \times 10^{-4}$ |
| Amazon | ClusterTree | n/a | 126.3 | n/a | n/a | $1.1 \times 10^{-4}$ |
| ($N_{\text{leaf}} = 30,249$) | qTWD ($\lambda = 10^{-3}$) | n/a | 162.2 | 368.8 | 5.6 | $1.2 \times 10^{-4}$ |
| | cTWD ($\lambda = 10^{-3}$) | n/a | 126.3 | 366.8 | 12.9 | $1.2 \times 10^{-4}$ |

Table 4 shows the classification accuracies for each method. The proposed weighting-based approaches compare favorably with the WMD (Sinkhorn). In particular, the proposed method can significantly improve the ClusterTree cases, whereas the improvements with the vanilla QuadTree method are modest.

## 5.4 Computation and memory costs

Table 5 summarizes the computational costs for training and testing. For training, we measured the average computational cost using a Xeon CPU E5-2690 v4 (2.60 GHz) for computing the cost matrix and constructing

Table 6: Memory usage of each method for the test time. Note that we report the sizes of dense matrices because dense matrices are more suited for the GPU.

| Methods | Twitter ($N_{\text{leaf}} = 4,489$) | | BBCSport ($N_{\text{leaf}} = 10,103$) | | Amazon ($N_{\text{leaf}} = 30,249$) | |
|---|---|---|---|---|---|---|
| | Cost matrix | Tree matrix | Cost matrix | Tree matrix | Cost matrix | Tree matrix |
| Sinkhorn | 161.2 MB | n/a | 816.6 MB | n/a | 7320.0 MB | n/a |
| QuadTree | n/a | 184.8 MB | n/a | 929.7 MB | n/a | 8149.3 MB |
| ClusterTree | n/a | 203.4 MB | n/a | 982.6 MB | n/a | 8089.1 MB |
| qTWD ($\lambda = 10^{-3}$) | n/a | 184.8 MB | n/a | 929.7 MB | n/a | 8149.3 MB |
| cTWD ($\lambda = 10^{-3}$) | n/a | 203.4 MB | n/a | 982.6 MB | n/a | 8089.1 MB |

the tree[4], training set generation for weight estimation, and the Lasso optimization. The training set generation process includes the calculation of the cost matrix and $z$ vectors. For the test, we computed the average time consumption for comparing the entire document in the training set with one document for the test. As can be observed, training set generation is the most computationally expensive operation for our proposed method. The computational costs of computing the cost matrix and tree construction were comparable. In contrast, the computational cost of the Lasso optimization was small. Overall, the computational cost of training increased when the number of training vectors $N_{\text{leaf}}$ increased. In the test, the computational speed of the proposed method was several orders of magnitude faster with GPUs. It is important to note that the proposed approximation approach cannot be used to compute the exact 1-Wasserstein distance; the proposed approach may not be suitable for tasks in which the 1-Wasserstein distance should be computed exactly.

**Memory cost:** Table 6 shows the required memory space for each method. For the Sinkhorn algorithm, it needs to store the $N_{\text{leaf}} \times N_{\text{leaf}}$ dimensional cost matrix. Tree methods need to store the $N \times N_{\text{leaf}}$ matrix, where $N$ ($N > N_{\text{leaf}}$) is the number of entire nodes. Thus, if we store the matrices in a dense format, the required memory size of the tree method is slightly larger than that of the Sinkhorn algorithm. Note that we report the size of dense matrices because they are more suited for GPU computation. However, for CPUs, because the tree matrix $B$ can be efficiently stored in a sparse format, the memory space required for $B$ is negligible. For a sparse matrix, with ClusterTree, the sizes of $B$ for the Twitter, BBCsport, and Amazon datasets are 94.4 KB, 226.2KB, and 681.1KB, respectively. Thus, if we compute the TWD with a CPU, the required memory space is much lower than that of the Sinkhorn-based algorithms.

## 6 Conclusion

In this study, we considered approximating the 1-Wasserstein distance with trees. More specifically, we first revealed that the 1-Wasserstein distance approximation can be formulated as a distance approximation problem. Then, we proposed a non-negative Lasso-based optimization technique for learning the weights of a tree. As the proposed method is convex, a globally optimal solution can be obtained. Moreover, we proposed a weight estimation procedure for a tree-sliced variant of the Wasserstein distance. Through experiments, we show that the proposed weight estimation method can significantly improve the approximation performance of the 1-Wasserstein distance for both the QuadTree and ClusterTree cases. Owing to L1-regularization, we can compress the tree size without losing the approximation performance.

A potential limitation of our proposal is that the tree-based approach may not be suitable for tasks that require exact evaluation of the 1-Wasserstein distance. Moreover, because a tree must be constructed from data, the proposed approach may not be suitable for the end-to-end training of deep neural network models. Another limitation of the tree-based approach is that it may not be suitable for matching problems, such as a semantic correspondence detection task (Liu et al., 2020). In contrast, the tree-based approach is very fast, with a reasonable approximation performance for the original 1-Wasserstein distance. Thus, the

---

[4]Due to our implementation of tree construction, the computational complexities of QuadTree and ClusterTree were $O(dN_{\text{leaf}})$ and $O(dN_{\text{leaf}}HK)$, respectively.

proposed tree-based approach is more suited for computing a large number of 1-Wasserstein distances, such as document retrieval and document classification tasks (i.e., $k$-nearest neighbor classifier).

## Acknowledgement

M.Y. was supported by MEXT KAKENHI Grant Number 20H04243. R.S. was supported by JSPS KAKENHI Grant Number 21J22490.

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

## A Proof of Proposition 1

Let $\mathbf{\Pi}^* = \mathbf{\Pi}^*(\mu, \nu)$. If there exists $(i, j, k)$ such that $\pi_{ij} > 0$ and $\pi_{jk} > 0$,

$$\pi'_{pq} = \begin{cases} \pi_{ij} - \min(\pi_{ij}, \pi_{jk}) & (p, q) = (i, j) \\ \pi_{jk} - \min(\pi_{ij}, \pi_{jk}) & (p, q) = (j, k) \\ \pi_{ik} + \min(\pi_{ij}, \pi_{jk}) & (p, q) = (i, k) \\ \pi_{pq} & \text{otherwise} \end{cases} \tag{19}$$

is no worse than $\mathbf{\Pi}^*$ because $d(i, j) + d(j, k) \geq d(i, k)$ is due to the triangle inequality. We assume no tuples $(i, j, k)$ exist such that $\pi_{ij} > 0$ and $\pi_{jk} > 0$ without loss of generality. If there exist $i_1, i_2, \cdots, i_{2n}, i_{2n+1} = i_1$ such that $\boldsymbol{P}_{i_1 i_2} > 0, \boldsymbol{P}_{i_3 i_2} > 0, \boldsymbol{P}_{i_3 i_4} > 0, \cdots, \boldsymbol{P}_{i_1 i_{2n}} > 0$, either

$$\pi'_{pq} = \begin{cases} \pi_{i_{2k+1} i_{2k+2}} - \varepsilon & (p, q) = (i_{2k+1}, i_{2k+2}) \\ \pi_{i_{2k+3} i_{2k+2}} + \varepsilon & (p, q) = (i_{2k+3}, i_{2k+2}) \\ \pi_{pq} & \text{otherwise} \end{cases} \tag{20}$$

or

$$\pi''_{pq} = \begin{cases} \pi_{i_{2k+1} i_{2k+2}} + \varepsilon & (p, q) = (i_{2k+1}, i_{2k+2}) \\ \pi_{i_{2k+3} i_{2k+2}} - \varepsilon & (p, q) = (i_{2k+3}, i_{2k+2}) \\ \pi_{pq} & \text{otherwise} \end{cases} \tag{21}$$

is no worse than $\mathbf{\Pi}^*$, where $\varepsilon = \min\{\pi_{i_{2k+1} i_{2k+2}} \mid k = 0, \cdots, n-1\} \cup \{\pi_{i_{2k+3} i_{2k+1}} \mid k = 0, \cdots, n-1\}$, and both supports of $\pi'_{pq}$ and $\pi''_{pq}$ do not have this cycle. Therefore, we assume the support of $\boldsymbol{P}^*$ has no cycle without loss of generality; that is, the graph is a tree. Let $\mathcal{T}$ be the graph induced by the support of $\mathbf{\Pi}^*$, and let the weight of edge $e = (i, j)$ be $w_e = d(w_i, w_j)$. As the shortest path distance between $u$ and $v$ on tree $\mathcal{T}$ is no less than $d(u, v)$ by definition, $W_1(\mu, \nu) \leq W_{\mathcal{T}}(\mu, \nu)$ holds. However, any pair $(i, j)$ on the support of $\mathbf{\Pi}^*$ is directly connected in $\mathcal{T}$. Therefore,

$$\begin{aligned} W_1(\mu, \nu) &= \min_{\mathbf{\Pi} \in U(\mu, \nu)} \sum_{i=1}^{n} \sum_{j=1}^{n'} \pi_{ij} d(\boldsymbol{x}_i, \boldsymbol{x}'_j), \\ &= \sum_{i=1}^{n} \sum_{j=1}^{n'} \pi^*_{ij} d(\boldsymbol{x}_i, \boldsymbol{x}'_j) \\ &= \sum_{i=1}^{n} \sum_{j=1}^{n'} \pi^*_{ij} d_{\mathcal{T}}(\boldsymbol{x}_i, \boldsymbol{x}'_j) \\ &\geq \min_{\mathbf{\Pi} \in U(\mu, \nu)} \sum_{i=1}^{n} \sum_{j=1}^{n'} \pi_{ij} d_{\mathcal{T}}(\boldsymbol{x}_i, \boldsymbol{x}'_j) \\ &= W_{\mathcal{T}}(\mu, \nu). \end{aligned}$$

Furthermore, $W_1(\mu, \nu) = W_{\mathcal{T}}(\mu, \nu)$ and $\boldsymbol{P}^*$ is an optimal solution for both $W_1(\mu, \nu)$ and $W_{\mathcal{T}}(\mu, \nu)$.

