# OpenReview forum: "Approximating 1-Wasserstein Distance with Trees"
_TMLR — Accepted by TMLR_

### Review · Reviewer_gte7 · 2022-07-02

**Summary Of Contributions:**

This paper correctly points out that existing tree-based approximations of Wasserstein distance, while computationally efficient, fail to satisfactorily approximate the vanilla Wasserstein distance. The authors propose to train the edge weights of the tree via a sparse lasso algorithm via an efficient algorithm that enables the resulting tree Wasserstein distance to closely approximate its vanilla counterpart (at least for the L1 norm).

**Broader Impact Concerns:**

None that I can think of.


**Requested Changes:**

Given the purpose of the paper is to introduce efficient approximation algorithm, I would like to see runtimes of the methods whose performance is reported in the Experiments.

I'd also like to see the authors discuss the extra memory / space requirement of maintaining the tree structure and weighted edges, which is not present in the vanilla Wasserstein distance calculation.

When listing the computational complexity of the proposed algorithm, clarify how is this affected by the dimensionality d of the datapoints x.

Given the popularity of the Sinkhorn algorithm, I would like to see it included in the empirical comparisons. It will help readers understand the pros/cons of the proposed approach vs the most popular approximate-Wasserstein algorithm that I'm aware of.


Here is another paper that could be discussed which efficiently approximates Wasserstein distance in 1D like max-sliced Wasserstein distance: "Principal Differences Analysis: Interpretable Characterization of Differences between Distributions"

Authors should comment why the focus is on 1-Wasserstein distance. I presume it is because trees are particularly well-suited for it but would be helpful to explicitly describe the intuition here.


**Strengths And Weaknesses:**

The paper is tackling a high impact problem (efficient computation of 1-Wasserstein distance), the contribution is original as far as I know, and the resulting algorithm is easily implemented (via off-the-shelf solver) and provably attains a global optimum. The paper is well-written overall with clear explanations/figures. The experiments seem thorough and correctly executed, with convincing results as to the practical utility of the proposed method.

I think this is a strong paper overall, the minor weaknesses I found are listed under requested changes.

---

> ### Author Response · Authors · 2022-07-31
> **Response to Reviewer gte7**
>
> Thank you for your valuable comments.
>
> > Given the purpose of the paper is to introduce efficient approximation algorithm, I would like to see runtimes of the methods whose performance is reported in the Experiments.
>
>  We have added the runtime experiments in Table 3. For constructing the tree, the most time-consuming part is to generate the training set for Lasso. However, for our experiment, we can construct a tree of about 10 min for the Amazon dataset. For the test, all tree-based approaches can be computed very fast by GPU (several orders of magnitude faster than the Sinkhorn algorithm).
>
> > I'd also like to see the authors discuss the extra memory / space requirement of maintaining the tree structure and weighted edges, which is not present in the vanilla Wasserstein distance calculation.
>
> In Table 4, we summarized the memory requirement of the tree structure in dense format. If we store them in dense format, the required memory space is comparable to storing the cost matrix of the vanilla Wasserstein distance. Since the tree matrix is sparse, the required memory size can be dramatically compressed if we store them in a sparse format.
>
> > When listing the computational complexity of the proposed algorithm, clarify how is this affected by the dimensionality d of the datapoints x.
>
>  It affects when we construct a tree. For the QuadTree case, the construction time is $O(d N_\text{leaf} \log 1/\epsilon)$, where $\epsilon$ is an accuracy parameter. For the ClusterTree case, it is $O(d N_\text{leaf} H \log K)$, where $H$ is the depth of tree and $K$ is the number of clusters if we use farthest-point clustering.  Note that, due to our implementation of tree-construction, the computational complexities of QuadTree and ClusterTree in this paper are $O(d N_{\text{leaf}})$ and $O(d N_{\text{leaf}} H K)$, respectively. In our experiments, we used $d = 300$, which is common for word vectors, and the computation of the QuadTree and the ClusterTree are reasonable (See Table 3.)
>
> > Given the popularity of the Sinkhorn algorithm, I would like to see it included in the empirical comparisons. It will help readers understand the pros/cons of the proposed approach vs the most popular approximate-Wasserstein algorithm that I'm aware of.
>
>  Thank you for the suggestion. We added the experiments with the Sinkhorn algorithm, whose results are shown in Table 2. For our experiments in document classification, our proposed method can get the comparable performance of the Sinkhorn algorithm. So, we observed that the tree Wasserstein could be highly useful for document comparison tasks. One of the potential limitations of our proposed tree-based algorithm is that we may not be able to obtain a good transport plan since we are approximating 1-Wasserstein distance with tree metric. Thus, we added a discussion about potential limitations in the conclusion section.
>
> > Here is another paper that could be discussed which efficiently approximates Wasserstein distance in 1D like max-sliced Wasserstein distance: "Principal Differences Analysis: Interpretable Characterization of Differences between Distributions"
>
>  Thank you for the information. Indeed the work is an important one. It is a max-sliced Wasserstein distance, and it is highly related to the subspace robust Wasserstein distance. We have added the reference in the paper.
>
>  We have not considered the max-sliced version of tree Wasserstein distance in our work. It is potentially an interesting future work.
>
> > Authors should comment why the focus is on 1-Wasserstein distance. I presume it is because trees are particularly well-suited for it but would be helpful to explicitly describe the intuition here.
>
>  Tree-Wasserstein distance is a 1-Wasserstein distance with tree-metric, so we focused on the 1-Wasserstein distance in this paper. It is not clear for other cases such as 2-Wasserstein distance. It may be possible to fit the squared distances by our proposed tree approach and approximate the 2-Wasserstein distance by 1-Wasserstein distance. It is indeed an exciting research direction, and we would like to consider it in the future.

---

### Review · Reviewer_JU6a · 2022-07-15

**Summary Of Contributions:**

Wasserstein distance is an important metric for many tasks such as document classification or retrieval. However, the computation of Wasserstein distance is computationally expensive. The prior works have discussed using tree-Wasserstein which can be evaluated faster than Wasserstein.
This paper proposed a method for learning the edge weights of the tree-Wasserstein such that it closely mimics the 1-Wasserstein distance.
The proposed method is based on Proposition 3 in the paper which shows
$$
d_\mathcal{T} (x_i, x_j) = \mathbf{w}^T (b_i + b_j -2b_i \circ b_j),
$$
where $b_i$ indicated the ancestors of vector $i$ in the tree.
Then they regress the edge weights $w$ such that  $d_\mathcal{T} (x_i, x_j)$ matches L2 distance between two vectors.
The closeness of the resulting $\mathcal{W}_{\mathcal{T}}$ to 1-Wasserstein is based on the previous work.


**Requested Changes:**

1- ClusterTree information is missing from Figure 3.d, e, c.
2- The training time for your algorithm should be presented in the paper.
3- The advantage of the proposed method in end tasks such as classification or document retrieval is unclear. The author should also study the performance of their method in, for example, document classification and compare the performance (accuracy) to just using QuadTree/ClusterTree and 1-Wasserstein. Is the difference significant enough to justify using the proposed method?
4- The author should also discuss the limitation of their work


Presentation:
1- Section 3.2: m and n' have been used interchangeably. Please use n' only to avoid confusion.
2- Section 3.3: if the node i is the parents node of the leaf node j -> if node i is the ancestor of leaf node j
3- Please add an equation number for the equations.


**Strengths And Weaknesses:**

Strength:
The paper is easy to read and follow.
The problem that they try to address is important to the community. The contribution is simple and straightforward.

Weakness:
The experimental studies are limited and do not show the effectiveness of their approach.

---

> ### Author Response · Authors · 2022-07-31
> **Response to Reviewer JU6a**
>
> Thank you for your valuable comments.
>
> > 1- ClusterTree information is missing from Figure 3.d, e, c. 2
>
>   The MAEs of ClusterTrees are more than 1, and thus they do not appear in the figure. We will add a note for this.
>
> > The training time for your algorithm should be presented in the paper.
>
>    We measure the training time and summarize it in Table 3. Moreover, we measure the required memory space for each method in Table 4.
>
> > The advantage of the proposed method in end tasks such as classification or document retrieval is unclear. The author should also study the performance of their method in, for example, document classification and compare the performance (accuracy) to just using QuadTree/ClusterTree and 1-Wasserstein. Is the difference significant enough to justify using the proposed method?
>
>  Thank you for the suggestion. We added the document classification experiments in Table 2. For this experiment, we found that the proposed weight estimation is particularly useful for the ClusterTree cases, while the performance gain in QuadTree is modest. It is important to note that the proposed weight method is helpful if the 1-Wasserstein distance is beneficial for the downstream tasks. So, if the performance of the 1-Wasserstein distance is not good, the performance of the proposed method can also be poor.
>
> > The author should also discuss the limitation of their work
>
> A potential limitation of our proposal is that our tree-based approach may not be suited for tasks that require the exact evaluation of 1-Wasserstein distance. Moreover, since it needs to construct a tree from data, the proposed approach may not be suited for the end-to-end training of deep neural network models. In contrast, the proposed approach is super fast with good approximation performance of the original 1-Wasserstein distance. Thus, the proposed approach is more suited for some NLP tasks, such as document retrieval and classification tasks.
>
> In the updated manuscript, these limitations are discussed in Section 6.
>
> > Presentation: 1- Section 3.2: m and n' have been used interchangeably. Please use n' only to avoid confusion. 2- Section 3.3: if the node i is the parents node of the leaf node j -> if node i is the ancestor of leaf node j 3- Please add an equation number for the equations.
>
>  Thank you for pointing out these issues. We fixed them and added the equation numbers.

---

### Review · Reviewer_GEEH · 2022-07-17

**Summary Of Contributions:**

The paper proposes an extension of Tree-Sliced Wasserstein and supervised Tree Wasserstein distances that propose to compute an approximation of Wasserstein distances by finding tree metrics related to the two distributions to be compared. The novelty in this paper is that Authors propose to compute the weights of the tree branches by solving a lasso problem which enables
to (theoretically) get as close as possible to the original Wasserstein distance. Experiments on document comparisons are proposed, showing that the proposed distance behaves similarly to the true Wasserstein distance.

**Requested Changes:**

Minor remarks/suggestion of improvments
  - instead of (Kolouri et al., 2016), [1] should be credited for the first paper on sliced Wasserstein distance. Also, what do you mean by ‘performances are highly dependent on the projection matrix’ ? This point should really be clarified
 - Section 3.1 and 3.2 are redundant. Please merge them.
 - after Prop 2., the sentence starting with ‘This proposition indicates …’ misses a verb
- Computational complexity: this part should definitely be enhanced with O() complexity for each step of the corresponding procedures-
- Related works: there are many other applications in ML of Wasserstein distances. I suggest authors complete this aspects of the paper with references (including for instance, generative modeling, domain adaptation or zero shot learning, inverse optimal transport, etc.). Also, Numerous papers use the concept of mini-batches to lower the computational burden of solving a full optimal transport problem by averaging over smaller (and more tractable) problems. This line of works should also definitely be cited and compared to.


[1] Julien Rabin, Gabriel Peyré, Julie Delon, and Marc Bernot. Wasserstein barycenter and its application to
texture mixing. In International Conference on Scale Space and Variational Methods in Computer Vision,
pp. 435–446. Springer, 2011

**Strengths And Weaknesses:**

While the formulation of the problem is interesting, and has potential, I think that in the current form the paper is missing key experiments showing the execution time benefits other regular solvers (EMD, sinkhorn or sliced versions). It is definitely not clear if the combination of the time needed to construct the tree, solve its weights with the Lasso solver, and compute the corresponding tree Wasserstein metric is faster on the considered problems.
Performances should also be reported for down stream tasks (in a context of word mover distances or other, I would suggest domain adaptation or more simply color adaptation). Notably, is there a direct correlation between low MAE and high performances ?

---

> ### Author Response · Authors · 2022-07-31
> **Response to Reviewer GEEH**
>
> Thank you for your valuable comments.
>
> > I think that in the current form the paper is missing key experiments showing the execution time benefits other regular solvers (EMD, sinkhorn or sliced versions).It is definitely not clear if the combination of the time needed to construct the tree, solve its weights with the Lasso solver, and compute the corr esponding tree Wasserstein metric is faster on the considered problems. Performances should also be reported for down stream tasks (in a context of word mover distances or other, I would suggest domain adaptation or more simply color adaptation). Notably, is there a direct correlation between low MAE and high performances ?
>
>  We included the tree construction time and the weight estimation in Table 3. With this experiment, the computational cost for training is not that large. The most time-consuming part of the current implementation is the training data generation, which includes the cost function computation and the vector calculation (i.e., z). However, since we can easily parallelizable the process, we can quickly speed up using multi-core CPUs. For the test, the computational speed of tree methods is several orders of magnitude faster than that of the Sinkhorn-based algorithm.
>
>  In addition to the computational comparison, we evaluated our proposed method using the document classification task (see Table 2 in the revised paper). Through this experiment, we observe that we can improve the classification accuracy over existing tree methods. In particular, for ClusterTree, the gain of the proposed weighting-based approach is significant, while the performance is comparable for the QuadTree case.
>
> > instead of (Kolouri et al., 2016), [1] should be credited for the first paper on sliced Wasserstein distance.
>
>  Thank you for pointing this out. We updated the reference information.
>
> > Also, what do you mean by ‘performances are highly dependent on the projection matrix’ ? This point should really be clarified
>
>  In the sliced Wasserstein, it first projects the input data into 1D space with a projection matrix. Thus, the performance changes if we change a projection matrix. We usually average the sliced Wasserstein distances with different projection matrices to avoid this problem. We clarified this in the updated paper.
>
> >  Section 3.1 and 3.2 are redundant. Please merge them.
>
>  We merged Section 3.1 and 3.2 into one subsection. Note that, in OT, we can set any cost function $c(x,x')$, while we can only use distance metric $d(x,x')$ for Wasserstein distance. So, we would like to keep both cases for readers.
>
> > after Prop 2., the sentence starting with ‘This proposition indicates …’ misses a verb
>
> "indicates" is a verb. To avoid the confusion, we added more information.
>
> This proposition indicates that the 1-Wasserstein distance with an arbitrary distance metric and TWD are equivalent (i.e., $W_1(\mu,\nu) = W_{\mathcal{T}}(\mu,\nu)$) if $d(x,x') = d_{\mathcal{T}}(x,x')$}
>
> > Computational complexity: this part should be enhanced with O() complexity for each step of the corresponding procedures-
>
>
> We added the computational complexity of the proposed method in Section 4.3. Moreover, we calculate the actual computational time for computing the cost matrix, tree construction, training set generation, and the Lasso optimization, as shown in Table 3.
>
> > Related works: there are many other applications in ML of Wasserstein distances.  I suggest authors complete this aspects of the paper with references (including for instance, generative modeling, domain adaptation or zero shot learning, inverse optimal transport, etc.). Also, Numerous papers use the concept of mini-batches to lower the computational burden of solving a full optimal transport problem by averaging over smaller (and more tractable) problems. This line of works should also definitely be cited and compared to.
>
> We added a more detailed discussion of the Wasserstein distance application in Section 2.

---

### Decision · Action_Editors · 2022-08-26

**Recommendation:** Accept with minor revision

**Comment:**

The paper proposes to approximate the 1-Wasserstein distance by the tree-Wasserstein distance by computing tree embeddings of the distributions involved and showing how a simple Lasso can retrieve the edge weights of the trees, which are needed to compute the distance approximation via a shortest path problem reduction.

The reviewers recognised the importance of approximations of this kind and the main results in the paper. During the review and discussion phase some concerns were raised about the time and space complexity of the proposed approximations as well as the limited experiments run. Authors are addressed these concerns in a satisfactory way, adding one more baseline, two downstream tasks and reporting the time and memory taken to build the trees and compute the distance sub-routines.

Even if in some scenarios the proposed methods might be slower than a baseline or require additional memory, I deem this work to be interesting enough to inspire follow up works that can close the gap for faster approximations. The paper is therefore accepted with minor revision. Authors are encouraged to incorporate the following minor and very minor changes in the camera ready.

Minor changes:
  - discuss the time complexity of all sub-routines. Stating that FISTA has an "efficient solver" is not clear. Discuss the
  - discuss the space complexity of the method and its competitors (e.g. Sinkhorn).
  - report error bars for metrics in Table 1 and 2

Very minor changes:
  - increase font sizes of axis labels in plots (e.g. Fig3)
  - draw tables without vertical bars and boxes (one can use the `booktabs`package, see eg, [1])

[1] https://nhigham.com/2019/11/19/better-latex-tables-with-booktabs/